:̈P̈LOS | ONE

# The impact of deceased donor maintenance on delayed kidney allograft function: A machine learning analysis

**Silvana Daher Costa**[1,2,3], **Luis Gustavo Modelli de Andrade**[4], **Francisco Victor Carvalho Barroso**[1], **Cláudia Maria Costa de Oliveira**[2,3], **Elizabeth De Francesco Daher**[1], **Paula Frassinetti Castelo Branco Camurça Fernandes**[2], **Ronaldo de Matos Esmeraldo**[3], **Tainá Veras de Sandes-Freitas**[1,3]*

1 Department of Clinical Medicine, Faculty of Medicine, Federal University of Ceará, Fortaleza, Ceará, Brazil, 2 Walter Cantídio University Hospital, Fortaleza, Ceará, Brazil, 3 Hospital Geral de Fortaleza, Fortaleza, Ceará, Brazil, 4 Department of Internal Medicine, Universidade Estadual Paulista—UNESP, Botucatu, São Paulo, Brazil

* taina.sandes@gmail.com

**Data Availability Statement:** Data supporting the findings of this study are available as supplementary file

## Abstract

### Background

This study evaluated the risk factors for delayed graft function (DGF) in a country where its incidence is high, detailing donor maintenance-related (DMR) variables and using machine learning (ML) methods beyond the traditional regression-based models.

### Methods

A total of 443 brain dead deceased donor kidney transplants (KT) from two Brazilian centers were retrospectively analyzed and the following DMR were evaluated using predictive modeling: arterial blood gas pH, serum sodium, blood glucose, urine output, mean arterial pressure, vasopressors use, and reversed cardiac arrest.

### Results

Most patients (95.7%) received kidneys from standard criteria donors. The incidence of DGF was 53%. In multivariable logistic regression analysis, DMR variables did not impact on DGF occurrence. In post-hoc analysis including only KT with cold ischemia time<21h (n = 220), urine output in 24h prior to recovery surgery (OR = 0.639, 95%CI 0.444–0.919) and serum sodium (OR = 1.030, 95%CI 1.052–1.379) were risk factors for DGF. Using elastic net regularized regression model and ML analysis (decision tree, neural network and support vector machine), urine output and other DMR variables emerged as DGF predictors: mean arterial pressure, $\geq 1$ or high dose vasopressors and blood glucose.

### Conclusions

Some DMR variables were associated with DGF, suggesting a potential impact of variables reflecting poor clinical and hemodynamic status on the incidence of DGF.

**Funding:** SDC received support for travel expenses from Fundação Cearense de Apoio ao Desenvolvimento Científico e Tecnológico (Funcap) and Coordenação de Aperfeiçoamento de Pessoal de Nível Superior - Brasil (CAPES). The funders had no role in study design, data collection and analysis, decision to publish, or preparation of the manuscript. Other authors received no specific funding for this work.

**Competing interests:** The authors have declared that no competing interests exist.

**Abbreviations:** AUC, Area under the receiver operating curve; BMI, Body Mass Index; CIT, Cold ischemia time; CPK, Creatine phosphokinase; DGF, Delayed graft function; DMG, Donor maintenance goals; DMR, Donor maintenance-related; DT, Decision tree; HLA, Human leucocyte antigen; HRSA, Health Resources and Services Administration; HTK, Histidine-tryptophan-ketoglutarate; IGL-1, Institut Georges Lopez-1; IRB, Institutional Review Board; KDPI, Kidney Donor Profile Index; KNN, K-nearest neighbors; KT, Kidney transplant; ML, Machine learning; MM, Mismatches; Na+, Sodium; NN, Neural Network; OPTN, Organ Procurement and Transplant Network; OR, Odds-ratio; PRA, Panel reactive antibodies; SCr, Serum creatinine; SVM, Support Vector Machine; UNOS, United Network for Organ Sharing; UW, University of Wiscosin.

## Introduction

Brazilian studies have reported incidences of delayed graft function (DGF) between 50 and 70%, 2 to 3-fold higher than the rates described by American and European cohorts, despite similar or more favorable recipient and donor demographics [1–6]. A Brazilian study reported 22.7% incidence of delayed kidney function in a cohort of simultaneous pancreas-kidney transplants, despite a short mean cold ischemia time of 14h and the use of ideal donors [7]. With similar demographics, international cohorts reported incidences of 4–5% [8, 9]. Notable, similar to demonstrated in American and European cohorts, DGF in Brazilian transplant recipients has negative impact on short and long-term outcomes [4, 6, 10].

There is no robust evidence explaining the high DGF incidence in our country, but it is likely that the suboptimal maintenance care of potential donors before organ recovery has an important role. Of note, recent studies have shown that achieving optimal donor maintenance parameters is associated with significant decrease in DGF occurrence [11, 12].

Traditionally, studies evaluating risk factors for DGF adopt standard statistical approaches, such as logistic regression. These models are useful in analysis using few independent variables, mainly when the effect of the predictor on the outcome is linear and homogeneous. The assumptions required to regression-based models are often not reached in clinical research and important predictor variables may be obscured. Machine learning (ML) methods can improve precision and accuracy in predicting events, by using more sensitive statistical methods, with data mining techniques and complex data interactions modeling non-linear interactions [13–15]. As an example, Decruyenaere et al. demonstrated that logistic regression was not the ideal method for DGF prediction in a Belgian cohort and ML methods performed better discriminative capacity [16]. Additionally, other regression-based models are useful and present better performance, depending on the number of events, number of predictors, variables characteristic and distribution [17].

This study aimed to evaluate the risk factors for DGF, including in the analysis donor maintenance-related (DMR) variables, which were thoroughly investigated from multidisciplinary records. To increase the analysis accuracy and properly investigate the impact of donor maintenance on DGF occurrence, we selected a cohort of brain dead donor (DBD) kidney transplants (KT) performed in a Brazilian region where DGF incidence is high despite the predominance of ideal donors. In addition, we used ML methods for data analysis beyond regression models.

## Materials and methods

### Study design

This study is a retrospective analysis from all deceased donor KT recipients older than 16 years of age, performed between January 1st 2015 and December 31st 2017 at two Brazilian transplant centers, located in a region with locally predominant use of standard criteria donors [18]. Preemptive, multiorgan transplants, recipients of machine perfused grafts and those who lost their grafts or died within 7 days after KT were excluded. In compliance with Brazilian law, all donors were brain dead.

Data were retrospectively collected by systematic review of medical charts and electronic database. Patient records and information was anonymized and de-identified prior to analysis. Due to the observational and retrospective nature of the study, with data anonymously analyzed, informed consent was not obtained.

The study was performed in accordance with ethical standards of National Health Council Resolution 466/12 and Declaration of Helsinki, and was approved by Institutional Review

Board (IRB) of the Federal University of Ceará (Ethics Committee approval number: 2.004.286) and by the IRBs of all hospitals involved in the donation and transplantation processes: Walter Cantídio University Hospital (2.183.661), Instituto José Frota (2.183.661) and Hospital Geral de Fortaleza (2.059.876).

### Definitions

Delayed graft function was defined as the requirement for at least one dialysis session during the first week after KT, regardless of the clinical indication [19]. DGF duration was assessed by the time until the last dialysis session, and by the number of sessions performed in this period.

To better test the hypothesis that variables not included in traditional DGF predictive models could explain the high incidence in our country, we opted to calculate the expected incidence of DGF using the nomogram described by Irish et al [1] as a starting point of the study. Among all available DGF predictive models, Irish nomogram has demonstrated the best predictive power in validation studies including non-American patients [20, 21]. This nomogram was developed using United Network for Organ Sharing / Organ Procurement and Transplantation Network (UNOS/OPTN) database and include the following variables: recipient ethnicity, gender, body mass index (BMI), and history of previous KT, diabetes or blood transfusion; time on dialysis, peak panel reactive antibodies (PRA), human leucocyte antigens mismatches (HLA MM); donor age, weight, history of hypertension and terminal serum creatinine (sCr), cause of death, donation after cardiac death, cold ischemia time (CIT), and warm ischemia time [1]. Since Irish nomogram does not allow the inclusion of machine-perfused kidneys, we opted to exclude them.

Donor maintenance parameters were evaluated using the Donor Management Goals (DMG) previously described by US Department of Health and Human Services, Health Resources and Services Administration (HRSA) [11, 22], with adaptations considering local peculiarities and the retrospective nature of the study: the lowest arterial blood gas pH during hospitalization was captured and patients who presented values between 7.3 and 7.45 were considered in the goal; the highest serum sodium ($Na^+$) during hospital stay was recorded and patients who presented values between 135 and 155 mEq/L were considered in the goal; the target for the highest blood glucose was $\leq$ 150 mg/dL; diuresis in the last 24h prior the recovery surgery was considered adequate when between 0.5 and 3 mL/Kg/h; the lowest mean arterial pressure was in the goal when between 60 and 110 mmHg; and target for vasopressors was the use of $\leq$ 1 vasoactive drug with norepinephrine <0.5 μg/Kg/min. The highest creatine phosphokinase (CPK), history of reversed cardiac arrest and acute kidney injury during hospitalization were also evaluated.

### Statistical analysis

Categorical variables were presented as frequency and percentage and compared using Chi-square or Fisher tests. Normally distributed continuous variables were summarized as mean and standard deviation and compared using Student's t-test. Median was included in description of non-parametric continuous variables and comparison was performed using Mann Whitney-test. A multivariable logistic regression model was fitted to compute covariate-adjusted odds ratios (OR) for DGF. Twenty-seven variables were included in the model. Collinear variables ("final sCr" and "difference between final and initial sCr") and those with more than 10% of missing values ("CPK") were excluded. Diabetic donors were also excluded, since this was a near-zero variance predictor (all patients who received grafts from diabetic donors developed DGF). A p-value of <0.15 in univariable analysis was considered statistically significant for including variables in multivariable analysis. For all other analysis, a p-value of

<0.05 was considered statistically significant. Statistical analysis was performed using SPSS v.23.0 software (SPSS, Inc., Chicago, IL, USA).

Predictive models for DGF were constructed using a supervised model, according to the following steps: data acquisition, data processing, model construction, and model evaluation. In data processing step, the outcome category was balanced to achieve equal proportions. In pre-process we perform recursive feature elimination to select the predictable variables with better correlation with the outcome. Variables with moderate / strong correlation with DGF ($r > 0.80$) were included in the model.

Population was randomly divided into training and testing set with a stratified 70:30 split. Six supervised ML algorithms were developed with this subset of variables in the training set: neural network (NN), support vector machine (SVM), C5 decision tree (DT), CHAID DT, k-nearest neighbors (KNN) and logistic regression with stepwise selection. Area under the receiver operating curve (AUC) was calculated to test the ability of each model to distinguish patients in testing set. SPSS Modeler v.18.1 (IBM, Armonk, NY, USA) was used to construct predictive models.

As an additional sensitivity analysis, we performed Elastic Net regression, a regularization model that mix Lasso and Ridge regressions. While Lasso regression enhances the prediction accuracy and interpretability of the statistical model by variable selection and regularization, Ridge regression improves prediction by shrinking large regression coefficients to reduce over-fitting. Elastic net regression also provides a more interpretable model when compared to black-box results of ML methods. For this analysis, numerical predictors were normalized by a Box and Cox transformation and after center and scale. Median were imputed for missing values. Categorical predictors were dummy encoded. We used an alpha of 0.1 and 10-fold cross-validation to search lambda. After an optimal search of a lambda, we perform an elastic net with R package glmnet [23]. Those coefficients inferior or greater than zero are considered relevant predictors. Variables whose coefficient was zero were considered not important for DGF prediction.

## Results

### Recipient and donor demographics

From 954 KT performed in the period, 30 were living donor transplants, 70 were allocated to patients younger than 16 years, 16 were multiorgan transplants, 8 were preemptive KT, 365 recipients received machine perfused grafts, and 22 lost the graft or died in the first week. The final analysis included 443 DD KT. The incidence of DGF was 53%, the mean time of DGF was 11.8 ± 15.0 days (median 7 days) and mean number of dialysis sessions was 5.0 ± 5.1 (median 4). According to Irish nomogram, the expected incidence of DGF was 19% and this tool showed poor predictive accuracy (AUC 0.685).

Recipient and donor demographic characteristics are detailed in Tables 1 and 2. Patients in DGF group were older (45.6 ± 14.4 vs. 42.6 ± 15.0 years old, p = 0.030), presented higher prevalence of pretransplant diabetes (21.3 vs. 13.5%, p = 0.034), and longer time on dialysis (36 vs. 27 months, p = 0.001). Donors in DGF group showed higher mean age (33.2 ± 12.6 vs. 28.5 ± 12.3 years old, p<0.001), higher body mass index (BMI) (25.7 ± 3.5 vs. 24.9 ± 4.0 Kg/m$^2$, p = 0.038), higher prevalence of hypertension (7.2 vs. 3.4%, p = 0.002), higher terminal sCr (1.2 ± 0.7 vs. 1.0 ± 0.5 mg/dL, p = 0.001), and higher Kidney Donor Profile Index (KDPI) (35.1 ± 23.0 vs. 28.4 ± 19.7%, p = 0.001). Of note, 433 donors (97.7%) presented KDPI ≤ 85%.

The main perfusion solution was histidine-tryptophan-ketoglutarate (HTK) (83.1%), followed by University of Wisconsin (UW) (13.1%) and Institute Georges Lopez-1 (IGL-1) (3.8%) and this variable did not impact on DGF incidence. There were no differences between

**Table 1. Recipient demographic and clinical characteristics.**

| | Total N = 443 | Without DGF (n = 208) | DGF (n = 235) | P value |
|---|---|---|---|---|
| **Gender–male** | 251 (56.7) | 121 (58.2) | 130 (55.3) | 0.545 |
| **Age** (yo) | 44.2 ± 14.7 | 42.6 ± 15.0 | 45.6 ± 14.4 | 0.030 |
| **Ethnicity** | | | | 0.126 |
| *Caucasian / white* | 35 (7.9) | 22 (10.6) | 13 (5.5) | |
| *Mixed race / hispanic* | 374 (84.4) | 169 (81.2) | 205 (87.2) | |
| *Afro-Brazilian / black* | 34 (7.7) | 17 (8.2) | 17 (7.2) | |
| **BMI** (Kg/m$^2$) | 24.3 ± 4.5 | 23.9 ± 4.3 | 24.7 ± 4.6 | 0.054 |
| **CKD etiology** | | | | 0.034 |
| *Unknown* | 134 (30.2) | 63 (30.3) | 71 (30.2) | |
| *GN* | 111 (25.1) | 66 (31.7) | 45 (19.1) | |
| *Diabetes* | 73 (16.5) | 26 (12.5) | 47 (20.0) | |
| *Hypertension* | 54 (12.2) | 24 (11.5) | 30 (12.8) | |
| *PKD* | 31 (7.0) | 10 (4.8) | 21 (8.9) | |
| *Urological* | 26 (5.9) | 12 (5.8) | 14 (6.0) | |
| *Other* | 14 (3.2) | 7 (3.4) | 7 (3.0) | |
| **History of diabetes** | 78 (17.6) | 28 (13.5) | 50 (21.3) | 0.034 |
| **Time on dialysis** (mo) | 46.8 ± 45.2 (34) | 40.3 ± 40.9 (27) | 52.5 ± 48.1 (36) | 0.001 |
| **Retransplantation** | 36 (8.1) | 15 (7.2) | 21 (8.9) | 0.602 |
| **Class I PRA** (%) | 9.9 ± 23.4 (0) | 9.4 ± 22.6 (0) | 10.2 ± 24.0 (0) | 0.766 |
| **Class II PRA** (%) | 4.2 ± 14.4 (0) | 4.0 ± 14.5 (0) | 4.3 ± 14.3 (0) | 0.633 |
| **HLA MM** | 3.6 ± 1.2 | 3.5 ± 1.3 | 3.6 ± 1.2 | 0.526 |
| **DSA**[1] | 27 (6.1) | 12 (5.8) | 15 (6.4) | 0.844 |

yo; years old; BMI; body mass index; CKD: chronic kidney disease; PKD: polycystic kidney disease; GN: glomerulonephritis; mo: months; PRA: panel reactive antibodies; HLA MM: human leucocyte antigen mismatches; DSA: donor specific anti-HLA antibodies.

[1] single-antigen bead assays (LabScreen Single Antigen; One Lambda) on Luminex platform with reactions showing mean intensity fluorescence > 1500.

groups regarding vascular anastomosis time (36.5 ± 12.3 vs. 35.6 ± 11.1 min, p = 0.440) and anti-thymoglobulin induction therapy (99.1 vs. 96.6%, p = 0.090). However, DGF group presented longer CIT (21.7 ± 3.8 vs. 20.1 ± 4.1h, p<0.001).

## Donor maintenance

Donor maintenance parameters data (Table 3) showed the poor clinical and hemodynamic conditions experienced by donors: high need for vasopressors (95.9%), substantial incidence of reversed cardiac arrest episodes (12.2%), high CPK (median 951 UI/L), increased serum Na$^+$ (160.0 ± 13.8 mEq/L) and poor blood glucose control (193.3 ± 77.7 mg/dL). DGF group had lower percentage of donors reaching blood glucose target (26 vs. 37.5%, p = 0.010) and lower urine output in the last 24h prior to recovery surgery (median 0.9 vs. 1.2 mL/Kg/h, p = 0.005).

## Risk factors for DGF

In logistic regression analysis, variables independently associated with DGF were: recipient history of diabetes (OR 1.922, 95% CI 1.119–3.302, p = 0.018), time on dialysis (OR 1.009, 95% CI 1.004–1.014, p<0.001), donor hypertension (OR 2.331, 95% CI 1.247–4.355, p = 0.008), final sCr (OR 1.947, 95% CI 1.320–2.872, p = 0.001), and CIT (OR 1.115 95% CI 1.058–1.175, p<0.001). However, logistic regression presented poor predictive performance, both in

**Table 2. Donor demographic and clinical characteristics.**

| | Total N = 443 | Without DGF (n = 208) | DGF (n = 235) | P value |
|---|---|---|---|---|
| **Age** (yo) | 31.0 ± 12.7 | 28.5 ± 12.3 | 33.2 ± 12.6 | < 0.001 |
| **BMI** (Kg/m$^2$) | 25.3 ± 3.8 | 24.9 ± 4.0 | 25.7 ± 3.5 | 0.038 |
| **Ethnicity** | | | | 0.714 |
| *Caucasian / white* | 51 (11.5) | 26 (12.5) | 25 (10.6) | |
| *Mixed race / hispanic* | 377 (85.1) | 174 (83.7) | 203 (86.4) | |
| *Afro-Brazilian / black* | 15 (3.4) | 8 (3.8) | 7 (3.0) | |
| **Hypertension** | 24 (5.4) | 7 (3.4) | 17 (7.2) | 0.002 |
| **Diabetes** | 3 (0.7) | 0 (0) | 3 (1.3) | 0.251 |
| **Brain death cause** | | | | 0.448 |
| *Trauma* | 314 (70.9) | 151 (72.6) | 163 (69.4) | |
| *Vascular event* | 96 (21.7) | 40 (19.2) | 56 (23.8) | |
| *Anoxia* | 24 (5.4) | 11 (5.3) | 13 (5.5) | |
| *Other* | 9 (2.0) | 6 (2.9) | 3 (1.3) | |
| **HCV** | 0 (0) | 0 (0) | 0 (0) | NA |
| **Final sCR**[1] (mg/dL) | 1.1 ± 0.6 | 1.0 ± 0.5 | 1.2 ± 0.7 | 0.001 |
| **ECD**[2] | 19 (4.3) | 5 (2.4) | 14 (6.0) | 0.065 |
| **KDPI** (%) | 31.9 ± 21.8 | 28.4 ± 19.7 | 35.1 ± 23.0 | 0.001 |
| **KDRI** | 0.86 ± 0.2 | 0.82 ± 0.2 | 0.89 ± 0.22 | 0.001 |

yo: years old; BMI: body mass index; HCV: hepatitis C virus; sCr: serum creatinine; na: not applicable; ECD: expanded criteria donor; KDPI: Kidney Donor Profile Index; KDRI: Kidney Donor Risk Index.

[1] last serum creatinine before harvest surgery.

[2] United Network for Organ Sharing (UNOS) definition: a) donors >60 yr of age or b) donos 50–59 yr of age with at least two of the following: sCr>1.5 md/dL, history of hypertension or cardiovascular death.

**Table 3. Donor maintenance parameters.**

| | Total N = 443 | Without DGF (n = 208) | DGF (n = 235) | P value |
|---|---|---|---|---|
| **Time to BD (days)** | 4.3 ± 4.2 (3) | 4.3 ± 3.6 (3) | 4.2 ± 4.7 (2) | 0.219 |
| **Reversed cardiac arrest** | 54 (12.2) | 30 (14.4) | 24 (10.2) | 0.192 |
| **Δ sCr (mg/dL)** | 0.2 ± 0.7 (0) | 0.1 ± 0.6 (0) | 0.2 ± 0.7 (0.1) | 0.135 |
| **CPK (IU/L)**[1] | 2554 ± 5256 (951) | 2161 ± 4876 (880) | 2903 ± 5561 (952) | 0.470 |
| **Arterial blood gas pH** | 7.33 ± 0.08 | 7.32 ± 0.08 | 7.32 ± 0.08 | 0.210 |
| **Arterial blood gas pH 7.3–7.45** | 233 (52.6) | 114 (54.8) | 119 (50.6) | 0.392 |
| **Serum Na$^+$ (mEq/L)** | 160.6 ± 13.8 | 159.6 ± 13.3 | 161.6 ± 14.2 | 0.127 |
| **Serum Na$^+$ 135–155 mEq/L** | 170 (38.4) | 82 (39.4) | 88 (37.4) | 0.696 |
| **Blood glucose (mg/dL)** | 193.3 ± 77.7 | 189.1 ± 81.0 | 197.1 ± 74.6 | 0.282 |
| **Blood glucose ≤150mg/dL** | 139 (31.4) | 78 (37.5) | 61 (26.0) | 0.010 |
| **Urine output (mL/Kg/h)** | 1.5 ± 1.7 (1.1) | 1.7 ± 2.2 (1.2) | 1.3 ± 1.0 (0.9) | 0.005 |
| **Urine output 0.5–3 mL/Kg/h** | 363 (81.9) | 172 (82.7) | 191 (81.3) | 0.712 |
| **Mean arterial pressure (mmHg)** | 83.0 ± 14.5 | 82.1 ± 15.0 | 83.7 ± 14.0 | 0.257 |
| **Mean arterial pressure 60–110 mmHg** | 385 (86.9) | 179 (86.1) | 206 (87.7) | 0.673 |
| **Vasopressors** | 425 (95.9) | 202 (97.1) | 223 (94.9) | 0.335 |
| **≤ 1 vasopressor and low dose** | 353 (79.7) | 164 (78.8) | 189 (80.4) | 0.723 |

time do BD: time since the hospitalization to brain death; Δ sCr: difference between terminal creatinine (immediately prior to recovery surgery) and the initial creatinine (at hospital admission); CPK: creatine phosphokinase; Na+: sodium.

training (AUC 0.686) and testing set (AUC 0.695). To best explore the impact of donor maintenance variables on DGF incidence, logistic regression multivariable analysis was repeated in two subgroups of CIT, based on the median value of the total cohort (21h). The incidences of DGF were 48.2% and 57.8% in CIT<21h and CIT≥21h subgroups, respectively. Of note, in CIT<21h subgroup, in addition to the previously demonstrated variables (time on dialysis, donor hypertension and CIT), urine output (OR 0.639, 95% CI 0.444–0.919, p = 0.016) and serum Na+ (OR 1.030, 95% CI 1.007–1.053, p = 0.010) were also risk factors for DGF (Table 4).

The results of Elastic Net regression are illustrated in Fig 1. Variables in red were risk factors for DGF and variables in blue were associated with reduced risk of DGF. This model presented better predictive performance when compared to logistic regression (AUC 0.749).

The three statistical methods of better performance in analyzes using ML techniques were: boosted DT using C5.0 algorithm (AUC 0.791), boosting NN (AUC 0.886), and SVM with polynomial kernel (AUC 0.784). Fig 2 illustrates ML results and predictive performance. Variable are presented using the feature importance graph, in which higher the value of the feature/variable, more important it was to predict DGF. The feature importance was normalized between 0 and 1 by dividing by the sum of all feature importance values. As demonstrated in Fig 2, urine output, mean arterial pressure, and the use of more than one vasoactive drug or high dose vasopressor were predictive of DGF in all models. Blood glucose was predictive of DGF in DT and SVM models.

## Discussion

This study suggest that poor donor clinical and hemodynamic status may impact on DGF occurrence, and this might explain the high incidence of DGF in Brazil, where the incidence is significantly higher than that the predicted by available formulas.

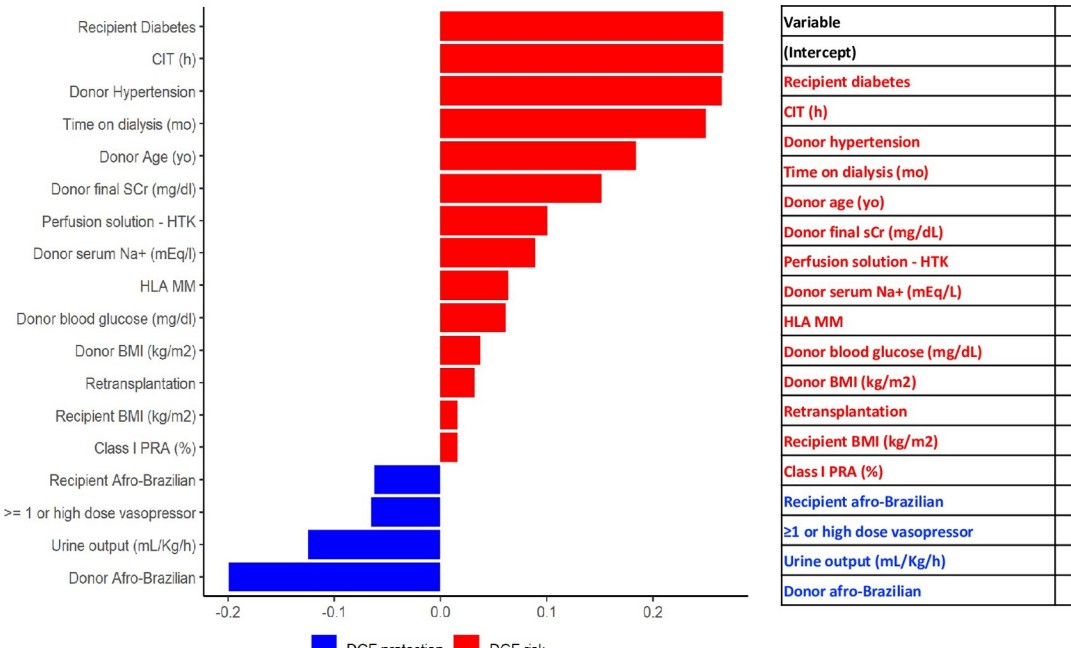

| Variable | Coefficients | OR |
|---|---|---|
| (Intercept) | 0.0066056 | 1.0066274 |
| Recipient diabetes | 0.2660105 | 1.3047488 |
| CIT (h) | 0.2659893 | 1.3047212 |
| Donor hypertension | 0.2650458 | 1.3034907 |
| Time on dialysis (mo) | 0.2496714 | 1.2836036 |
| Donor age (yo) | 0.1839115 | 1.2019094 |
| Donor final sCr (mg/dL) | 0.1514257 | 1.1634919 |
| Perfusion solution - HTK | 0.1008238 | 1.1060817 |
| Donor serum Na+ (mEq/L) | 0.0890321 | 1.0931158 |
| HLA MM | 0.0637048 | 1.0657777 |
| Donor blood glucose (mg/dL) | 0.0612088 | 1.0631209 |
| Donor BMI (kg/m2) | 0.0372825 | 1.0379862 |
| Retransplantation | 0.0325051 | 1.0330392 |
| Recipient BMI (kg/m2) | 0.0163103 | 1.0164440 |
| Class I PRA (%) | 0.0162292 | 1.0163616 |
| Recipient afro-Brazilian | -0.0625985 | 0.9393205 |
| ≥1 or high dose vasopressor | -0.0652805 | 0.9368047 |
| Urine output (mL/Kg/h) | -0.1247193 | 0.8827447 |
| Donor afro-Brazilian | -0.1995530 | 0.8190968 |

**Fig 1. Elastic net regression.** Variable in red (coefficient > zero) are risk factors for DGF and variables in blue (coefficient < zero) are protective. Variables are disposed by importance. DGF: delayed graft function; CIT: cold ischemia time; yo: years old; BMI: body mass index, PRA: panel reactive antibodies; HLA MM: human leucocyte antigen mismatches; sCr: final serum creatinine; Na+: sodium; VAT: vascular anastomosis time.

**Table 4. Logistic regression analysis of risk factors for DGF, according to CIT groups.**

| | | Total N = 443 | | | CIT < 21h N = 220 | | | CIT ≥ 21h N = 223 | | |
|---|---|---|---|---|---|---|---|---|---|---|
| | | OR | CI 95% | P value | OR | CI 95% | P value | OR | CI 95% | P value |
| Recipient demographics | Age (yo) | 0.994 | 0.979–1.010 | 0.492 | | NS | | 1.009 | 0.988–1.031 | 0.404 |
| | Afro-Brazilian | 1.237 | 0.726–2.109 | 0.434 | | NS | | 1.623 | 0.753–3.497 | 0.216 |
| | BMI (Kg/m$^2$) | 1.020 | 0.970–1.073 | 0.433 | | NS | | | NS | |
| | Diabetes | **1.922** | **1.119–3.302** | **0.018** | 1.471 | 0.681–3.178 | 0.327 | 1.043 | 0.977–1.113 | 0.209 |
| | Time on dialysis (mo) | **1.009** | **1.004–1.014** | **<0.001** | **1.007** | **1.001–1.013** | **0.021** | **1.012** | **1.003–1.020** | **0.008** |
| | Retransplantation | | NS | | | NS | | | NS | |
| | Class I PRA (%) | | NS | | | NS | | | NS | |
| | Class II PRA (%) | | NS | | | NS | | | NS | |
| | HLA MM | | NS | | | NS | | | NS | |
| | DSA | | NS | | | NS | | | NS | |
| Donor demographics | Age (yo) | 1.016 | 0.999–1.034 | 0.066 | 1.021 | 0.995–1.048 | 0.113 | 1.010 | 0.983–1.037 | 0.486 |
| | BMI (Kg/m$^2$) | 1.041 | 0.979–1.106 | 0.204 | 1.030 | 0.933–1.137 | 0.561 | | NS | |
| | Afro-Brazilian | | NS | | | NS | | | NS | |
| | Hypertension | **2.331** | **1.247–4.355** | **0.008** | **2.751** | **1.215–6.232** | **0.015** | **2.588** | **1.112–6.027** | **0.027** |
| | Cerebrovascular death | | NS | | | NS | | | NS | |
| | Final sCr (mg/dL) | **1.947** | **1.320–2.872** | **0.001** | 1.714 | 0.895–3.283 | 0.104 | **1.803** | **1.093–2.972** | **0.021** |
| Donor maintenance | Reversed Cardiac arrest | | NS | | | NS | | | NS | |
| | Urine output (mL/Kg/h) | 0.926 | 0.761–1.126 | 0.442 | **0.639** | **0.444–0.919** | **0.016** | 1.054 | 0.835–1.330 | 0.660 |
| | Serum Na+ (mEq/L) | 1.011 | 0.996–1.026 | 0.155 | **1.030** | **1.007–1.053** | **0.010** | | NS | |
| | ≥ 1 or high dose vasopressor | | NS | | | NS | | | NS | |
| | Blood glucose (mg/dL) | | NS | | | NS | | | NS | |
| | Mean arterial pressure (mmHg) | | NS | | | NS | | | NS | |
| | Arterial blood gas pH | | NS | | | NS | | | NS | |
| Other | Perfusion solution–HTK | | NS | | | NS | | | NS | |
| | CIT (h) | **1.115** | **1.058–1.175** | **<0.001** | **1.205** | **1.052–1.379** | **0.007** | **1.179** | **1.037–1.341** | **0.012** |
| | VAT (min) | | NS | | | NS | | | NS | |
| | rATG induction | 3.046 | 0.536–17.311 | 0.209 | | NS | | | NS | |

DGF: delayed graft function; CIT: cold ischemia time; yo: years old; BMI: body mass index, PRA: panel reactive antibodies; HLA MM: human leucocyte antigen mismatches; D.S.A: donor specific antibodies; sCr: final serum creatinine; Na+: sodium; VAT: vascular anastomosis time; rATG: rabbit anti-thymocyte globulin. NS: Variables not included in multivariable model since p value >0.15 in univariable analysis.

In our cohort, DGF incidence was almost 3-fold higher than the predicted by the nomogram described by Irish et al, suggesting an important role of variables not included in the prediction model. In fact, none of the available predictive models includes in final formula variables reflecting donor maintenance. Except for terminal serum creatinine (that could reflect renal consequences of hypovolemia, shock and other causes of acute kidney injury), only the score developed by Chapal et al analyzed some variable related to donor care (type of vasopressor) [24].

Only 4.3% of patients received kidneys from expanded criteria donors and 97.7% had KDPI below 85%, suggesting good structural quality kidneys. On the other hand, 12.2% had a reversed cardiac arrest episode before organ recovery surgery, 47.4% presented acid-base disorders, 61.6% had hypo- or hypernatremia, and 68.6% showed inadequate glycemic control, reflecting poor clinical and hemodynamic conditions. The impact of poor donor maintenance in our country has previously suggested in a study including simultaneous pancreas-kidney transplants. In this cohort, despite favorable demographics, the incidence of delayed kidney

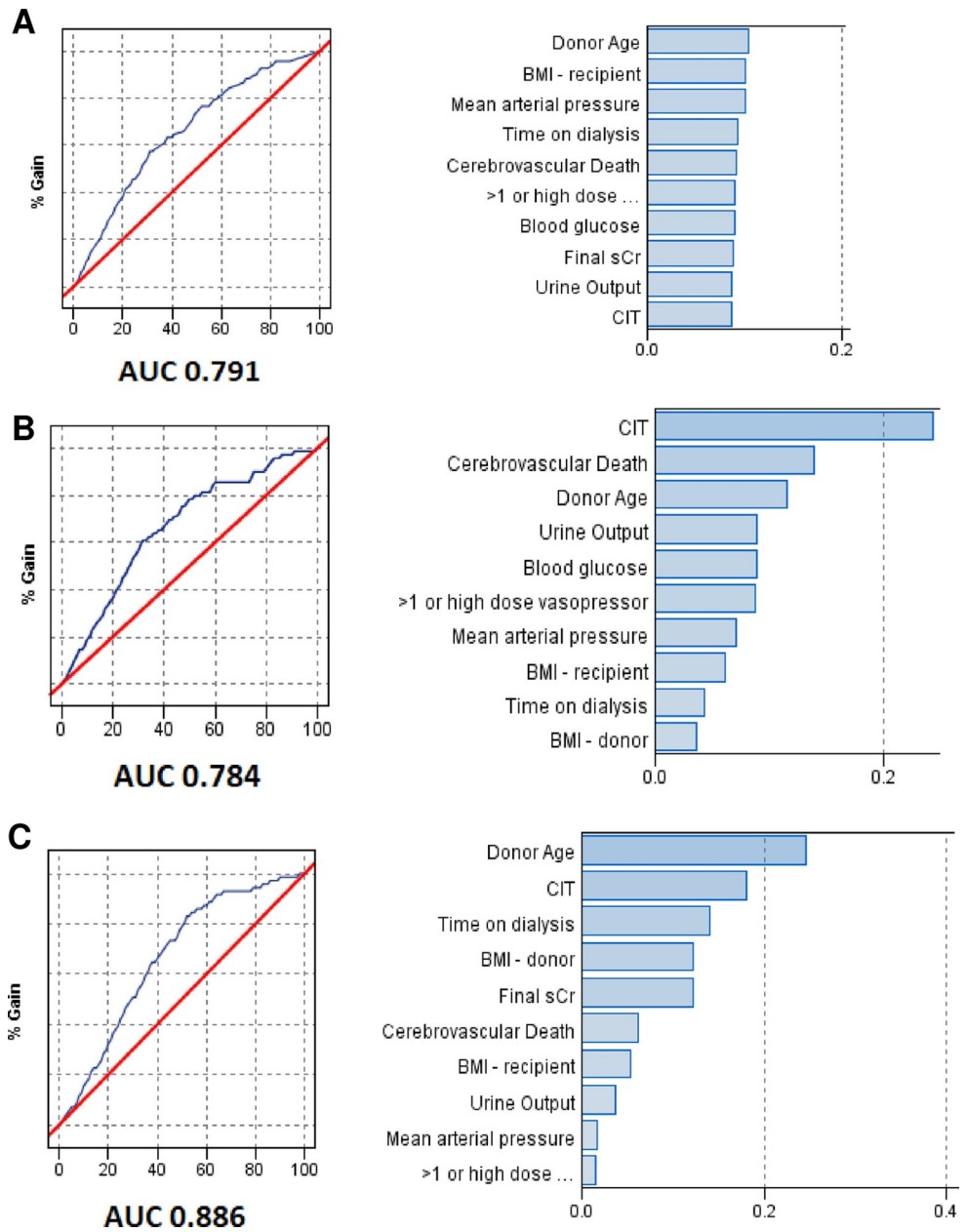

**Fig 2. Figs on the left demonstrate the area under the receiver-operating curve (AUC) for each method in the testing set.** Figs on the right show global variable importance for prediction of delayed graft function. (A) Decision Tree; (B) Support Vector Machine; (C) Neural Network. BMI: body mass index; sCr: final serum creatinine; CIT: cold ischemia time.

graft function was 22.7% and donor hypernatremia was an independent risk factor for DGF [7].

The challenge of properly maintaining potential deceased donors seems not to be exclusive of our population. Previous American and Canadian studies have reported low adherence to donor-care bundles at the time of consent for donation. Importantly, the early achievement of donor management goals was associated with a reduced risk of DGF in these studies [11, 12].

The high CIT was notable and its contribution to DGF is unequivocal. All statistical analysis demonstrated that each additional hour matters, even in KT with CIT < 21h. The large

territorial extension, the allocation model predominantly based on HLA compatibility, and the absence of specific allocation policies for "marginal" donors contribute to the long CIT in our country [25]. However, it is noteworthy that almost half of KT with CIT <21h presented DGF, suggesting the contribution of other factors beyond the CIT.

Due to the high negative impact of CIT on DGF incidence, potentially masking other predictors, we performed a post-hoc analysis including a regression model on a sub-sample of patients with CIT inferior to 21h. As hypothesized, variables reflecting donor maintenance now emerged. Additionally, in ML analysis, donor maintenance related variables, such as blood pressure, use of high dose vasopressors, urine output and blood glucose, were also associated with DGF. To further explore the contribution of other than the traditional variables to DGF incidence using a more interpretable model, we performed a sensitivity analysis using elastic net regression. Again, beyond the traditional conditions associated to DGF occurrence, variables reflecting donor management were risk factors (serum Na+, blood glucose) or protective (high dose vasopressors and diuresis).

Recently published studies demonstrated the impact of donor hemodynamics as a predictor of DGF in transplantation from donors after cardiac death [26, 27]. However, evidences are scarce on transplantation of brain dead donors.

Our study has limitations that should be pointed out. First, it was a retrospective cohort; therefore, the capture of variables was limited to those available on medical records. We cannot assure that our results are generalizable to other transplant centers worldwide. Donor maintenance related variables (and DMGs) are not static and we could not evaluate them dynamically. Since we could not follow variables over time and assess adherence to care bundles, it is possible that some clinical and hemodynamic data reflected the severity of the disease that led to brain death and not the patient care. Besides, it's possible that some clinical and hemodynamic parameters reflected patient situation before they became a consented donor, and thus may not reflect donor maintenance, but general intensive care. Finally, variables reflecting perioperative care were not available.

In conclusion, DGF incidence in Brazil is significantly higher than that predicted by available models. Although our data do not allow us to draw definitive conclusions, this study suggests that donor illness severity and hemodynamic instability might contribute to this scenario. Prospective studies are needed to robustly conclude how donor management impacts on DGF incidence. Additionally, a cohort including machine-perfused grafts may be useful to explore if pumping might mitigate kidney damage secondary to poor donor clinical and hemodynamic status.

We believe bringing this issue up is crucial in our setting. Scant donor care is probably a reflection of the poor economic conditions of our country. However, some educational actions could be taken, focusing on early recognition of potential donors and training staff who care for brain death patients.

## Supporting information

**S1 Database.**
(XLSX)

## Author Contributions

**Conceptualization:** Silvana Daher Costa, Elizabeth De Francesco Daher, Tainá Veras de Sandes-Freitas.

**Data curation:** Silvana Daher Costa, Cláudia Maria Costa de Oliveira, Paula Frassinetti Castelo Branco Camurça Fernandes, Ronaldo de Matos Esmeraldo, Tainá Veras de Sandes-Freitas.

**Formal analysis:** Silvana Daher Costa, Luis Gustavo Modelli de Andrade, Tainá Veras de Sandes-Freitas.

**Investigation:** Silvana Daher Costa, Francisco Victor Carvalho Barroso, Tainá Veras de Sandes-Freitas.

**Methodology:** Silvana Daher Costa, Luis Gustavo Modelli de Andrade, Tainá Veras de Sandes-Freitas.

**Project administration:** Silvana Daher Costa, Tainá Veras de Sandes-Freitas.

**Resources:** Silvana Daher Costa, Luis Gustavo Modelli de Andrade, Francisco Victor Carvalho Barroso, Cláudia Maria Costa de Oliveira, Elizabeth De Francesco Daher, Paula Frassinetti Castelo Branco Camurça Fernandes, Ronaldo de Matos Esmeraldo, Tainá Veras de Sandes-Freitas.

**Software:** Luis Gustavo Modelli de Andrade, Tainá Veras de Sandes-Freitas.

**Supervision:** Tainá Veras de Sandes-Freitas.

**Validation:** Silvana Daher Costa, Tainá Veras de Sandes-Freitas.

**Visualization:** Silvana Daher Costa, Tainá Veras de Sandes-Freitas.

**Writing – original draft:** Silvana Daher Costa, Luis Gustavo Modelli de Andrade, Tainá Veras de Sandes-Freitas.

**Writing – review & editing:** Silvana Daher Costa, Luis Gustavo Modelli de Andrade, Francisco Victor Carvalho Barroso, Cláudia Maria Costa de Oliveira, Elizabeth De Francesco Daher, Paula Frassinetti Castelo Branco Camurça Fernandes, Ronaldo de Matos Esmeraldo, Tainá Veras de Sandes-Freitas.

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
