## [Decision Letter · Decision Letter 0]

7 Nov 2019

PONE-D-19-26074

The impact of deceased donor maintenance on delayed kidney allograft function: a machine learning analysis

PLOS ONE

Dear MD, PhS Sandes-Freitas,

Thank you for submitting your manuscript to PLOS ONE. After careful consideration, we feel that it has merit but does not fully meet PLOS ONE’s publication criteria as it currently stands. Therefore, we invite you to submit a revised version of the manuscript that addresses the points raised during the review process.

ACADEMIC EDITOR: 

Interesting MS on donor management and the impact on DGF in kidney transplantation. However, quite a few issues were raised by the reviewers on the design of the study and specific results/outcomes, overstated conclusions, etc, that would need to be dealt with in great detail in order for this paper to be publishable. Please be advised that re-submission does not guarantee acceptance.

We would appreciate receiving your revised manuscript by Dec 22 2019 11:59PM. To enhance the reproducibility of your results, we recommend that if applicable you deposit your laboratory protocols in protocols.io, where a protocol can be assigned its own identifier (DOI) such that it can be cited independently in the future. For instructions see: http://journals.plos.org/plosone/s/submission-guidelines#loc-laboratory-protocols

We look forward to receiving your revised manuscript.

Kind regards,

Frank JMF Dor, M.D., Ph.D., FEBS, FRCS

Academic Editor

PLOS ONE

Journal Requirements:

3. In ethics statement in the manuscript and in the online submission form, please provide additional information about the patient records used in your retrospective study. Specifically, please ensure that you have discussed whether all data were fully anonymized before you accessed them and/or whether the IRB or ethics committee waived the requirement for informed consent. If patients provided informed written consent to have data from their medical records used in research, please include this information.

Reviewers' comments:

Reviewer's Responses to Questions

**Comments to the Author**

1. Is the manuscript technically sound, and do the data support the conclusions?

Reviewer #1: Yes

Reviewer #2: Partly

2. Has the statistical analysis been performed appropriately and rigorously? 

Reviewer #1: I Don't Know

Reviewer #2: I Don't Know

3. Have the authors made all data underlying the findings in their manuscript fully available?

Reviewer #1: Yes

Reviewer #2: Yes

4. Is the manuscript presented in an intelligible fashion and written in standard English?

Reviewer #1: Yes

Reviewer #2: Yes

5. Review Comments to the Author

Reviewer #1: Costa et al have used machine learning to assess the effects of donor management on DGF and compared that to predictive figures from a multivariate algorithm. The study is clearly written and has a fair sample size. The approach using ML is new and inventive. The evaluation of the exact mathematical analysis is not my expertise.

I have some reservations though on the design of the study.

1. The focus of the study is a bit unclear to me. Since the title and introduction are on machine learning, I dont know why first a predicitive model is used to show the higher than expected incidence of DGF. Even if DGF is in the expected incidence a correlation with donor factors could be found.

2. Data are very specific to countries or situations in which the DGF rate is this exceptionally high (for DBD). The mean KIT is very high and I dont think any European center will accept a KIT >21h for 50% of the transplants.The high DGF rate is cause by the long KIT untill proven otherwise, and not by donor management.

3. Therefore I would like to question how sure you are that you are actually relating to donor management issues, since these donors are brain dead and dying. Are the factors analyzed, e.g. the need for inotropics or urine output, a measure for the quality of donor mamangement or simply a refelection of the severity of the illness of the donor or the phase in the brain dead process the donor is in?

Reviewer #2: General Comments

• The rise of donation after circulatory death – where delayed graft function (DGF) in kidney transplants is common yet long term results are not inferior to donation after brain death, the end point of DGF is now considered much less important. That said, if donor maintenance can prevent DGF then that should of course be explored.

• I am unsure why DGF is so high in Brazil. You suggest because of poor donor management? Is there evidence of this?

• I am not sure your study does show, as you suggest it does, that poor donor maintenance impacts on DGF. You have shown that donors with certain physiological parameters are associated with DGF. But you have not shown that the cause of the poor physiological parameters is a result of poor donor maintenance. They just may be a sicker cohort. One with more hypoxic injury – as evidenced by being more likely to have had a cardiac arrest – probably from out of hospital cardiac arrest. So, I don’t accept your conclusion. To prove your conclusion, you would have to examine adherence to a donor care bundle, or match cohorts between hospitals that manage donors differently.

• Expanding this point, I would consider the following donor parameters may just reflect a sicker patient cohort: donor age, BMI, final sCr, cerebrovascular death, urine output, MAP, > 1 or high dose inotrope. Perhaps only blood glucose might be something the intensivist may have been able to control better.

• You describe in your methods ‘donor maintenance parameters’ but the results for which can be taken from anytime ‘during hospitalization’. So the worst result may be before the patient became a consented donor and thus may not reflect donor maintenance, but general intensive care.

• A shame the machine perfused kidney transplant cohort were not included. That would be interesting to see if machine perfusion can mitigate against poor donor maintenance.

• I was unclear what you hope can be done with your machine learning Figure 1? Is it useful because it describes the problem? Could it become a prediction tool – helping a transplant surgeon decide if a kidney needed machine perfusion?

Specific and Minor Comments

• You should say in your introduction (and ideally abstract) all were donors after brain death.

• ‘Due to the observational and retrospective nature of the study, with data anonymously analyzed, [the] informed consent was not obtained.’ No need for [the].

• Donor Maintenance. ‘poor hemodynamic conditions to which donors were submitted:’. This reads like someone submitted the donor to these problems, whereas they just may reflect the way things are and not negatively on donor maintenance. Example – reversed cardiac arrest. Was that cardiac arrest reversed in ICU. Or cardiac arrest reversed in the community leading to hypoxic brain injury and brain death. The first could represent poor donor maintenance. The second is juts the ways things are.

• Should be ‘The challenge of properly maintaining potential deceased donors seems not to be exclusive of our population.’

6. PLOS authors have the option to publish the peer review history of their article (what does this mean?). If published, this will include your full peer review and any attached files.

Reviewer #1: No

Reviewer #2: No

---

## [Author Response · Author response to Decision Letter 0]

9 Nov 2019

Dear Editor,

We are resubmitting our manuscript entitled “The impact of deceased donor maintenance on delayed kidney allograft function: a machine learning analysis" addressing all the issues raised by the reviewers (Rebuttal Letter attached). We appreciate all the comments, which contributed to the improvement of our manuscript, and thank you for the opportunity to resubmit this revised version.

Yours sincerely,

Tainá de Sandes-Freitas

---

## [Decision Letter · Decision Letter 1]

2 Dec 2019

PONE-D-19-26074R1

The impact of deceased donor maintenance on delayed kidney allograft function: a machine learning analysis

PLOS ONE

Dear MD, PhS Sandes-Freitas,

Thank you for submitting your manuscript to PLOS ONE. After careful consideration, we feel that it has merit but does not fully meet PLOS ONE’s publication criteria as it currently stands. Therefore, we invite you to submit a revised version of the manuscript that addresses the points raised during the review process.

ACADEMIC EDITOR: 

It would be very important that authors address the comments of reviewer 1 (and reviewer 2) in more detail, as there seems to be interest in the topic and the MS. However, reviewer 1 remains very critical about the explantations given so far. Also, the conclusions should probably be downsized significantly and limitations more clearly recognised by authors.

We would appreciate receiving your revised manuscript by Jan 16 2020 11:59PM. To enhance the reproducibility of your results, we recommend that if applicable you deposit your laboratory protocols in protocols.io, where a protocol can be assigned its own identifier (DOI) such that it can be cited independently in the future. For instructions see: http://journals.plos.org/plosone/s/submission-guidelines#loc-laboratory-protocols

We look forward to receiving your revised manuscript.

Kind regards,

Frank JMF Dor, M.D., Ph.D., FEBS, FRCS

Academic Editor

PLOS ONE

Reviewers' comments:

Reviewer's Responses to Questions

**Comments to the Author**

1. If the authors have adequately addressed your comments raised in a previous round of review and you feel that this manuscript is now acceptable for publication, you may indicate that here to bypass the “Comments to the Author” section, enter your conflict of interest statement in the “Confidential to Editor” section, and submit your "Accept" recommendation.

Reviewer #1: (No Response)

Reviewer #2: All comments have been addressed

2. Is the manuscript technically sound, and do the data support the conclusions?

Reviewer #1: Partly

Reviewer #2: Yes

3. Has the statistical analysis been performed appropriately and rigorously? 

Reviewer #1: I Don't Know

Reviewer #2: Yes

4. Have the authors made all data underlying the findings in their manuscript fully available?

Reviewer #1: Yes

Reviewer #2: Yes

5. Is the manuscript presented in an intelligible fashion and written in standard English?

Reviewer #1: Yes

Reviewer #2: Yes

6. Review Comments to the Author

Reviewer #1: I thank the author for the comments and explanations. They added explainig phrases to the manuscript. Unfortunately, the explanations the authors have given are not providing new insights. Moreover, it appears that the conclusions of the article are preliminary. For example, the contribution of KIT was not clarified, except for the statement that <21h was still 50% DGF. The contribution of donor ilness severity was only speculated on. No data were added to test the hypothesis that donor managment or treatment was indeed in a causative relation to DGF.

I think the manuscript has too many uncertainties and speculations to make it acceptable in this form.

Reviewer #2: General Comments

Very much improved paper.

Congratulations.

Specific and Minor Comments

In Statistical analysis, better: ‘Diabetic donors were also excluded, since this was a near-zero variance predictor (all patients who received grafts from diabetic donors developed DGF).’

In Results, better: ‘The mean time of DGF was 11.8 ± 15.0 days’

Conclusion perhaps could be: ‘Additionally, a cohort including machine-perfused grafts may be useful to explore if pumping might mitigate kidney damage secondary to poor clinical and hemodynamic status.’

This is probably enough given you didn’t prove causation to donor management even if suggestive.

7. PLOS authors have the option to publish the peer review history of their article (what does this mean?). If published, this will include your full peer review and any attached files.

Reviewer #1: No

Reviewer #2: No

---

## [Author Response · Author response to Decision Letter 1]

16 Jan 2020

We are resubmitting our manuscript entitled “The impact of deceased donor maintenance on delayed kidney allograft function: a machine learning analysis" addressing all the issues raised by the reviewers (attached). 

We appreciate all the comments, which contributed to the improvement of our manuscript, and thank you for the opportunity to resubmit this revised version.

Yours sincerely,

Tainá Veras de Sandes-Freitas

---

## [Editor Report · Decision Letter 2]

21 Jan 2020

The impact of deceased donor maintenance on delayed kidney allograft function: a machine learning analysis

PONE-D-19-26074R2

Dear Dr. Sandes-Freitas,

We are pleased to inform you that your manuscript has been judged scientifically suitable for publication and will be formally accepted for publication once it complies with all outstanding technical requirements.

With kind regards,

Frank JMF Dor, M.D., Ph.D., FEBS, FRCS

Academic Editor

PLOS ONE

Additional Editor Comments (optional):

The authors have now addressed all remaining issues raised by the reviewers. Happy to accept the 2nd revision.
---

## [Editor Report · Acceptance letter]

23 Jan 2020

PONE-D-19-26074R2 

The impact of deceased donor maintenance on delayed kidney allograft function: a machine learning analysis 

Dear Dr. Sandes-Freitas:

I am pleased to inform you that your manuscript has been deemed suitable for publication in PLOS ONE. Congratulations! Your manuscript is now with our production department. 

With kind regards,

on behalf of

Dr. Frank JMF Dor 

Academic Editor

PLOS ONE